# Natural Products for the Prevention, Treatment and Progression of Breast Cancer

**DOI:** 10.3390/cancers15112981

**Published:** 2023-05-30

**Authors:** Fabiano Svolacchia, Sergio Brongo, Alessia Catalano, Agostino Ceccarini, Lorenzo Svolacchia, Alessandro Santarsiere, Carmen Scieuzo, Rosanna Salvia, Francesca Finelli, Luigi Milella, Carmela Saturnino, Maria Stefania Sinicropi, Tommaso Fabrizio, Federica Giuzio

**Affiliations:** 1Department of Medical-Surgical Sciences and Biotechnologies, La Sapienza University, 00118 Rome, Italy; fabiano.svolacchia@gmail.com (F.S.); lorenzo.svolacchia@uniroma1.it (L.S.); 2Department of Medical Sciences, Policlinic Foundation Tor Vergata University, 00133 Rome, Italy; 3Department of Plastic Surgery, University of Salerno, 84131 Campania, Italy; sergiobrongo@gmail.com; 4Department of Pharmacy-Drug Sciences, University of Bari “Aldo Moro”, 70126 Bari, Italy; alessia.catalano@uniba.it; 5U.O.C. Primary Care and Territorial Health, Social and Health Department, State Hospital, 47893 San Marino, San Marino; agostino.ceccarini@iss.sm; 6Department of Science, University of Basilicata, 85100 Potenza, Italy; alessandro.santarsiere@unibas.it (A.S.); carmen.scieuzo@unibas.it (C.S.); r.salvia@unibas.it (R.S.); luigi.milella@unibas.it (L.M.); carmela.saturnino@unibas.it (C.S.); 7CNRS, UMR 7042-LIMA, ECPM, Université de Strasbourg, Université de Haute-Alsace, 67000 Strasbourg, France; 8Spinoff XFlies s.r.l., University of Basilicata, 85100 Potenza, Italy; 9Complex Structure UOP-AOSG “Moscati”, 83100 Avellino, Italy; medfinelli@gmail.com; 10Department of Pharmacy, Health and Nutritional Sciences, University of Calabria, 87036 Arcavacata di Rende, Italy; s.sinicropi@unical.it; 11Department of Plastic Surgery, IRCCS, Referral Cancer Center of Basilicata, 85028 Rionero in Vulture, Italy; tommaso.fabrizio@crob.it; 12Spinoff TNcKILLERS s.r.l., University of Basilicata, 85100 Potenza, Italy

**Keywords:** breast cancer, natural anticancer agents, chemoscience, prolonged inflammation

## Abstract

**Simple Summary:**

Cancer is one of the most dangerous diseases in humans, and no permanent therapy has been developed yet. Breast cancer (BC) is one of the most common cancers in women. The purpose of this review was to clarify how natural products may play a role in the prevention, treatment and progression of BC. For all the compounds examined, in vitro and in vivo studies, as well as clinical studies on BC, are described.

**Abstract:**

In this review, we summarize the most used natural products as useful adjuvants in BC by clarifying how these products may play a critical role in the prevention, treatment and progression of this disease. BC is the leading cancer, in terms of incidence, that affects women. The epidemiology and pathophysiology of BC were widely reported. Inflammation and cancer are known to influence each other in several tumors. In the case of BC, the inflammatory component precedes the development of the neoplasm through a slowly increasing and prolonged inflammation that also favors its growth. BC therapy involves a multidisciplinary approach comprising surgery, radiotherapy and chemotherapy. There are numerous observations that showed that the effects of some natural substances, which, in integration with the classic protocols, can be used not only for prevention or integration in order to prevent recurrences and induce a state of chemoquiescence but also as chemo- and radiosensitizers during classic therapy.

## 1. Introduction

The onset, maintenance and progression of BC are influenced by multiple pathophysiological aspects and signaling pathways [1,2]. Neoplastic phenomena recognize genetic and epigenetic components, and alterations of the differentiation program are often associated with the onset of neoplastic pathologies that involve the inhibition of the end of the replicative cycle in the differentiated cells and their permanence in the G0 phase [3]. Inflammation and cancer influence each other [4], and in some tumors, specifically BC, the inflammatory component precedes the development of the neoplasm, and a slowly increasing and prolonged inflammation linked to the neoplasm also favors tumor growth [5]. Several factors are related to cancer onset, progression and therapy. Many pathophysiological elements play important roles in the onset, maintenance and progression of BC, including inflammation, heat shock proteins (HSPs), matrix metalloproteinases (MMPs), immunological modulation, peritumoral inflammation, microRNAs and neoplastic stem cells, where the role of each of them appears to be linked to the others such that the signaling pathways that favor the BC are activated. In fact, the pro-inflammatory nuclear factor kB (NF-kB) is involved in cancer development [6]; cytokines, such as cyclooxygenase-2 (COX-2), epidermal growth factor (EGF) and vascular endothelial growth factor (VEGF), are the effectors of tumor tissue promotion [7]; HSPs, such as heat shock factor 1 (HFS-1) and hypoxia inducing factor (HIF-1), are widely studied in relation to BC [8]. Moreover, the involvement of reactive oxygen species (ROS) and superoxide dismutase (SOD) in BC were widely documented [9], as well MMPs [10], insulin-like growth factor (IGF) [11], fibroblast growth factors (FGFs) [12], microRNA [13] and so on. The mechanisms responsible for their involvement in cancer are briefly summarized here. Surgery, radiotherapy and chemotherapy are the main treatments for BC [14]. Moreover, nutritional intervention in BC patients is considered an integral part of the multimodal therapeutic approach [15]. However, most common therapies for BC lead to the occurrence of side effects and may also damage healthy tissues and organs. In this context, the use of natural agents integrated with conventional therapies can interact favorably with the onset of BC and classic surgical and medical therapy in order to reduce resistance to chemotherapy drugs, allow for better pharmacological aggression to neoplastic cells through chemosensitization and radiosensitization to allow for the chemoquiescence of neoplastic stem cells, which are insensitive to classic treatments [16]. Indeed, about 83% of the Food and Drug Administration (FDA)-approved chemotherapeutics are from natural sources [17]. Numerous natural products, such as meadowsweet, witch hazel, linseed, hops, Melissa leaves, Orthosiphon, Rosemary leaves, Goldenrod, green anise, Arnica flowers, Boldo leaves, Calendula, Echinacea root, Eleutherococcus, Humulone and feverfew, were examined for their generic antitumor characteristics and, in particular, their influence on BC [18]. Several plant-based anticancer compounds were studied for the treatment of cancer [19], and specifically for BC [20]. Our attention was focused particularly on curcumin, epigallocatechin gallate (EGCG), indole-3-carbinol (I3C), artesunate, ginger, flavonoids and bioflavonoids (xanthohumol and citrus bioflavonoids), myrrh, sulforaphane, vitamin D, medicinal mushrooms, acetylsalicylic acid and metformin for their direct influence on BC. In this review, we highlight and summarize these compounds and their interference in the onset, maintenance and progression of BC due to their influence on multiple pathophysiological and signaling aspects involved in this disease and we underline the usefulness of these compounds in the management of BC. 

## 2. Molecular Basis of BC 

The pro-inflammatory NF-kB is implicated in tissue inflammatory status, specifically in BC activation and progression through the stimulation of the cell proliferation, pro-survival and metastatic pathways of angiogenesis. Commonly used therapeutic agents, such as radiotherapy and chemotherapy, can also lead to the activation of NF-kB. This effect can contribute to the resistance of neoplastic cells to the agents themselves and the use of NF-kB inhibitors as sensitizers for therapy could improve the therapeutic target [21]. NF-kB is also regulated by a zinc finger deubiquitating regulatory enzyme A20/TNFaIP3 (tumor necrosis factor, alpha-induced protein 3), and new A20/TNFAIP3 functions were recently described, including the control of necroptosis, inflammasome activity [22] and the strong upregulation of the negative inflammatory regulator A20 (Tnfaip3), which is accompanied by a pronounced downregulation of the canonical NF-kB pathway [23]. Peritumor inflammation plays an important role in breast carcinogenesis. It allows BC cells to acquire the ability to survive, proliferate and disseminate. Inflammatory cells release some cytokines, such as COX-2, EGF and VEGF, which are the effectors of tumor tissue promotion; an alteration of immunity cells, both in number and in function, at sites of tumor progression is the generic constituent of the tumor microenvironment [24]. Moreover, HSPs play an important role in malignant transformation and progression by means of their intrinsic molecular chaperone properties that allow for the expression of new malignant traits through a facilitated accumulation of altered oncoproteins. In BC, the levels of HSPs are high. Their concentration in BC is directly related to the concentration of HFS-1, which then enables HSP transcription since HFS-1 inactivators are normally bound to HSPs [25]. Furthermore, the cells of the organism can be altered and undergo pathological phenomena when an excess of ROS comes into contact with the biological molecules that constitute them. Through an excessive presence of ROS, proteins, lipids and nucleic acids can be altered, but above all, there is an amplification of the inflammatory response and the induction of mutagenesis, promotion and neoplastic progression. ROS are able to damage lipids, nucleic acids and proteins with the alteration of their functional properties, and generally, the levels of enzymes that allow for antioxidant action are lower in cancer patients [26]. Through the ROS, the activation of signals is facilitated, which, through the intracellular Ca^++^, triggers the response of NF-kB. ROS are responsible for spontaneous mutations, where they can cause DNA breakage of 1 or 2 strands, deletions, insertions, loss of a base, substitution of a base and a deficiency of SOD in the mitochondria and cytoplasm of many solid neoplasms. The possibility of neoplastic cells exploiting all the resources contributes to growth and cachexia and also derives from the increase in a factor, namely, HIF-1, which increases if stimulated by growth factors [27]. MMPs are frequently overexpressed in neoplastic cells and are associated with a poorer prognosis. MMPs mediate the increase in growth factors, including IGF and FGF. There is a strong change in the tissue into which the tumor grows, especially in multifocal breast neoplasms. The growth factor IGF-1 strongly stimulates the production of aromatase, which is the major enzyme responsible for the synthesis of estradiol. Moreover, normal peritumoral cells cross-talk with tumor cells, supplying them with growth factors, such as EGF, IGF-2, FGF, transforming growth factor-beta (TGF-beta), human platelet-derived growth factor (PDGF) and VEGF, as well as regulating their motility. TGF-beta is a growth factor commonly secreted by tumor cells to evade immune surveillance and impair the regulation of cyclins [28]. An abnormal increase in cyclins affects the onset and maintenance of neoplastic cells. Cyclin A is inhibited by sulforaphane at dosages of 25 µM and by taxol (a commonly used chemotherapy sourced from the yew plant), which also inhibits cyclin B1. Cyclin D1 is inhibited by blocking NF-kB, which inhibits glycogen synthetase, which prevents its degradation; curcumin dephosphorylates retinoblastoma protein with the downregulation of D1 [29]. This dephosphorylation also inhibits cyclins E and A. Selectins with an EGF-like domain, such as L, P and E, are strongly implicated in BC. Selectin E is responsible for the reactivation of micrometastases even beyond 10 years, as well as the metastatic increase of the neoplasm. In addition, the receptors expressed on the membrane of the neoplastic cells, such as CD44^+^, are necessary for the process of bone invasion and cell survival. During the neoplastic transformation, the cells of breast tissue lose all the mechanism regulators of cell cycle control, such as P53, retinoblastoma protein (pRB) and P27. Moreover, two genes homologous to P53, namely, P63 and P73, may interact with P53. Cancer stem cells (CSCs) were proposed as one of the determining factors contributing to tumor heterogeneity [30]. They are insensitive or very insensitive to chemotherapy. Neoplastic stem cells are long-lived, almost perennial cells that have been exposed to true genotoxic stress more than their differentiated progeny, which, in any case, have a shorter lifespan. The view that neoplastic stem cells exist is related to two observations: heterogeneity and cell hierarchy. The first is based on the observation that almost all tumors can arise from a single cell, while within the tumor, the cells are not all the same; the second is that the tumor is able to self-regenerate through a small number of cells [31,32]. Within these cells, some signaling pathways are also deregulated, such as the Hedgehog, Wnt/β-catenin, Notch, phosphatase and tensin homologs, deleted on chromosome TEN (PTEN) and TGF-beta/BMP pathways. They can remain dormant for a long time and, luckily, they are very rare [31]. A microenvironment favorable to them promotes the maintenance of stem cells, which are essential for the regulation of homeostasis and contribute to tumorigenesis if altered during adulthood [33]. During experiments in guinea pigs, the neoplastic cells can only be transplanted into immunodeficient mice. The main modulators in the elimination of tumor cells are CD8+ lymphocytes and cytotoxic lymphocytes (CTLs) [34]. CTLs are obviously assisted by CD4+ lymphocytes and NK cells, and it has long been known that neoplastic growth is associated with functional alterations of the cytotoxic effectors and reduced NK-mediated cytotoxic activity. Some of the immunosuppressive factors of the immune system locally linked to the tumor are VEGF, TGF-beta and prostaglandin E2 [35]. There are short stretches of RNA in the cytoplasm, namely, microRNAs, that do not encode proteins, but the molecules produced by these short stretches are able to influence the regulation of gene expression contrary to the classic tumor suppressor or oncofacilitator genes. MicroRNAs can undergo binding with messenger RNAs; in this way, the blockage of a protein or its early degradation with the loss of function of that specific protein is determined. MicroRNAs function as oncogenes when their overexpression negatively regulates tumor suppressor genes or genes that control differentiation and apoptosis at the end of the cell life cycle. On the other hand, when microRNAs are underexpressed, they allow the blockade of a protein to be impaired and, therefore, function as oncofacilitators. The role of a microRNA depends on its target exclusively in that particular tissue and a decrease of tumor suppressor miRNAs is observed during neoplastic phenomena. The microRNA can, therefore, interfere with breast carcinogenesis. MicroRNA 30 A inhibits vimentin, while microRNA 31 is overexpressed in cell lines resistant to aromatase inhibitors that have strong oxidative stress [36]. The latter, in turn, inhibits metastasis in MDA-MD-231, while its downregulation allows for the migration of MCF-7 cells [37]. MicroRNA 34A is regulated by P53, and in BC, mutations in P53 can be observed and those present in P63 are resistant to contaxan chemotherapy and aromatase inhibitors. Through its transcription and presence, microRNA 644a allows for sensitization to drug therapy and the inhibition of the progression of the disease [38]. An overexpression or increase in oncofacilitator microRNAs, such as 15a/16-1 microRNAs, targets B-cell leukemia/lymphoma 2 protein (BCL2) messenger RNA that protects cells from apoptosis. Their down-expression by E2 transcription factor 7 (E2F7) enables cells to resist tamoxifen [39]. MicroRNAs 15/16/21 target PTEN, which encodes a phosphatase that transduces the phosphoinositide 3-kinase (PI3-kinase) signal. PTEN is mutated or deleted in BC [40]. MicroRNAs 221 and 222 regulate the protein P27, which inhibits the transition from the G1 phase to the S phase. MicroRNA 222 is down-expressed in aromatase inhibitor-resistant cell lines [36]. Preclinical studies on BC cell lines are often carried out on MCF-7 cells, which are the wild population of human BC cells. They possess histone deacetylase and are ER+. The P53 mutants are T47D (C44 and selectin). MDA-MB-231 cell lines belong to the highly invasive BC cell line (CERB+, C44+ and E+ selectin). MCF-10AT are the preneoplastic cells of the BC.

## 3. Polyphenols

Polyphenols are antioxidant substances capable of protecting cells from damage caused by free radicals [41]. Polyphenols present several therapeutic effects against different pathological states, including cancer; inflammation; diabetes; and heart, cardiovascular and related diseases [42]. Their involvement in the prevention and treatment of cancer was widely described [43,44]. The extraordinary ability of polyphenols to remit oxidative stress, lipid metabolism, insulin resistance and especially inflammation related to the onset of neoplastic diseases has attracted interest in clinical studies [45]. The mechanism of action for the anticancer effect of polyphenols is likely related to the modulation of microRNAs [46], mitogen-activated protein kinase (MAPK) signaling [47] and epigenetics [48]. Polyphenols may also affect the ability to overcome chemoresistance in cancer cells [49]. After decades in which polyphenols were considered highly safe compounds, nowadays, depending on the conditions, dose and interactions with the environment, the possibility for polyphenols to also exert harmful effects is now being investigated [50]. For example, in particular situations, the consequences of their ability to block iron uptake in some subpopulations can be harmful, as well as the possible inhibition of digestive enzymes, inhibition of intestinal microbiota, interactions of polyphenolic compounds with drugs and impact on hormonal balance [50]. 

### 3.1. Curcumin

Curcumin (Table 1), which is a polyphenolic compound, is an extract from *Curcuma longa* and is obtained via solvent extraction from the dried and ground rhizome of the *Curcuma longa* plant. Curcumin and its active metabolites are capable of directly binding to DNA and RNA, thereby sensitizing chemotherapeutics through the modulation of protein kinase C, telomerase, NF-kB and histone deacetylase (HDAC). Curcumin induces cell death and restores sensitivity to tamoxifen in antiestrogen-resistant BC cell lines [51]. From this, NF-kB, SRc and AKT/mTor are inactivated; enhancer of zeste 2 policomb repressive complex 2 subunit (EZH2) is epigenetically modified; cyclin-dependent kinase (CDK), P21, CIPI and P53 are upregulated; and transcription factors are inhibited and deactivated. Curcumin may regulate the gene expression of NF-kB, activator protein-1 (AP1), TNF, interleukin (IL), signal transducer and activator of transcription 3 (STAT3), peroxisome proliferator-activated receptor gamma (PPARγ), C-myc, BCL2, COX-2, nitric oxide synthase (NOS), cyclin D1, MMP-9, growth factors bFGF, EGF, granulocyte colony-stimulating factor (GCSF), IL-8, PDGF, TGF-alpha, VEGF, fibronectin and vibronectin [52,53]. Curcumin demonstrated inhibition of MCF-7 BC cell viability in a concentration-dependent manner and synergizes with mitomycin-c through p38 [54,55]. Moreover, curcumin increases the sensitivity of BC cells to the chemotherapeutics paclitaxel, cisplatin and doxorubicin [56,57]. It has antiproliferative effects on various hormone-dependent, independent and chemotherapy-resistant BC cell lines by inhibiting multi-drug resistance (MDR) [58] and confers a sensitizing effect during radiotherapy [59,60]. It induces concentration-dependent G2/M growth arrest [61]. Curcumin shows effects on miRNAs: it inhibits oncogenic ones, such as miR-21 and miR-186, and downregulates cyclin D1, cyclin E and mouse double minute 2 homolog (MDM2), while upregulating tumor suppressors, such as miR-15, miR-16, miR-22, retrovirus 27a, miR-34, miR-22, P21, p27 and p53. It has potential antiproliferative, anti-invasive and antiangiogenic effects as a mediator of chemoresistance and radioresistance. Curcumin is safe, even at 8 g per day for three months [62], enhances the antitumor activity of gemcitabine [63], inhibits E-selectin in vitro [64,65], inhibits HIF-1 [66] and Hedgehog signaling pathways [67,68], and regulates the P63 subfamily of the P53 family [69].

### 3.2. Epigallocatechin-3-Gallate

EGCG is an important polyphenolic component originating from green tea extract. It possesses various biological functions, including anti-cancer and anti-inflammatory properties [70]. EGCG was shown to enhance the expression levels of let-7, miR-15 and miR-16, and inhibits miR-20 and the progression of breast neoplasia [71]. The effects of EGCG on several types of MMPs were recently under study in the context of its anticancer activity [72]. Its importance as a cancer epigenetic regulator was recently reported [73]. It acts as a potential chemopreventive agent in BC [74]. Recently, its efficacy in preventing dermatitis in patients with BC was also highlighted [75]. The dose indicated by ESCOP ranges from 50 to 1600 mg (approximately 0.7–23 mg/kg body weight, based on a 70 kg body weight) [18].

## 4. Indole-3-Carbinol

I3C is a natural organic substance derived from the degradation of glucobrassicin glucosinolate, which is present in almost all cruciferous plants. Diindolylmethane is the biologically active derivative that spontaneously forms from I3C due to the action of gastric juices [76]. I3C is capable of limiting the proteolytic process of cyclin E in proto-oncogenes, inhibiting ERK in the signaling process of MCF/7 cells, inducing apoptosis in cells expressing EGFR and SRC, and eliminating the expression of IGF receptors [77]. It induces the overexpression of P21, P27 and GADD 457; translocation of FOX03A; and inhibition of AKT. It downregulates estrogen and downregulates cyclin E [78]. It induces apoptosis by externalizing membrane phosphatidylserine with DNA fragmentation and activation of caspase 3. It allows the upregulation of P21 and P27 and inhibition of CDK and their association with cyclins D1 and E. It inhibits the phosphorylation of pRB and downregulates NF-kB; it induces G1 phase arrest and the apoptosis of neoplastic cells [79]. It directs neoplastic stem cells toward differentiation [80]. It allows for the re-expression of microRNA31a [81] and interferes positively with aromatization [82]. It negatively influences the viability of ERa+ cells and synergizes the action of doxorubicin and cisplatin [83]. It negatively influences the viability of triple-negative cells [84]. It induces arrest in G2 cells and apoptosis in MCF7 and MB-MDA-231 cells [85]. It inhibits cyclin E, which is inhibited through I3C that limits the proteolytic process of cyclin E in proto-oncogenes, and C-Myc is the target of estrogen, which, when inhibited, reduces the expression of cyclin D1 [86] and arrests cells in a quiescent state. I3C is described as having significant safety and efficacy in biomedical fields [87]; its safety in normal cells was demonstrated [88]. It can be considered a safe natural substance from 300 mg to 600 mg/die [18]. 

## 5. Artesunate

Artesunate is a compound obtained from the plant derivative of Artemisia (*Artemisia annus*). It was found to have biological activity against a variety of cancers [89]. It is generally used in 600 mg/die [18]. Artesunate inhibits the expression of HSP70 and Bcl-2 in mammary BC 4T1 and MCF-7 cell lines [90]. It is able to regulate microRNA-34a, which is, in turn, regulated by P53 [91,92]. MicroRNA-34 was found to be unregulated in many human cancers and is considered a tumor suppressor microRNA due to its synergistic effect with the known tumor suppressor p53. miR-34 plays a key role in the inhibition of BC progression by not allowing the epithelial–mesenchymal transition (EMT) via the transcription factors’ EMT, p53 and some important signaling pathways [91]. Artesunate promotes mesenchymal/epithelial differentiation via the miR-34a pathway [92]. The lack of regulation of P53 via mutation on microRNA-31a confers resistance to taxane chemotherapy and aromatase inhibitors; therefore, the regulation of microRNA-31a positively influences chemoquiescence.

## 6. Ginger and Its Constituents

### 6.1. 6-Gingerol

6-Gingerol is a natural analog of curcumin derived from the root of ginger (*Zingiber officinale)* belonging to the Zingiberaceae family, which has a biological activity profile similar to that of curcumin [93]. Ginger is one of the oldest spices and contains bioactive compounds, among which 6-gingerol is the compound with the most interesting pharmacological properties [94]. 6-Gingerol may be extracted with conventional and nonconventional extraction techniques. Hydroalcoholic solutions and liquid CO_2_ are the most appropriate solvents for the extraction of 6-gingerol, whereas microwave-assisted extraction is the best extraction method [95,96]. The highest purity of 6-gingerol may be obtained with high-speed counter-current chromatography [97]. 6-Gingerol may also be obtained from other sources, including traditional Tibetan medicine *Dracocephalum heterophyllum* [98]. 6-Gingerol exerts its action through important mediators and cellular signaling pathways, including Bax/Bcl2, p38/MAPK, Nrf2, p65/NF-κB, TNF-α, ERK1/2, SAPK/JNK, ROS/NFκB/COX-2, caspase-3 and -9, and p53, and is able to provide antiproliferative, antitumor, antimetastatic and anti-inflammatory activities [99]. 6-Gingerol selectively kills BC stem cells (BCSCs) and increases sensitivity to paclitaxel, especially in those expressing CD44^+^ [100]. It is able to significantly decrease cell viability in a dose-dependent manner, particularly in osteosarcoma cells [101]. It sensitizes cell death induced by TRIAL (Tumor necrosis factor (TNF)-related apoptosis-inducing ligand) via apoptosis [102,103] and sensitizes the chemotherapy protocol with cyclophosphamide/adriamycin, preventing the initiation and progression of BC through inhibitory effects [104,105]. Despite the numerous biological properties of 6-gingerol, its low bioavailability is the main challenge that limits its application. Novel encapsulation and solubilization techniques, including nano-emulsion, complexation, micelles and solid dispersion methods, were introduced to enhance the bioavailability of 6-gingerol, overcoming its limitations [106]. To date, no side effects affecting any organ or body district have been reported. However, it was recently suggested to carry out clinical trials to further estimate the acute and chronic toxicity to assess the safety of ginger and related bioactive compounds [107]. In doses up 2 g/die os, 6-gingerol can be considered an absolutely safe natural substance [18]. 

### 6.2. 6-Shogaol (6SG)

6-Shogaol is a compound found in ginger and was shown to interact with a cellular receptor functioning as an antiproliferative, antimetastatic and proapoptotic agent [108,109,110]. A recent study reported that Notch signaling downregulation (Hes1 and CyclinD1 genes) caused by 6-shogaol leads to antiproliferative activity in breast cancer cells. Moreover, treatment with 6-shogaol induced significative and time-dependent cell cycle accumulation in G_2_/M-phase. It also induced significant apoptosis in BC cells and inhibited autophagy in BC cell lines, which might force these cells to undergo apoptosis [111].

## 7. Flavonoids and Bioflavonoids

### 7.1. Xanthohumol

Xanthohumol is a prenylated flavonoid derived from *Humulus lupulus* L. (hop plant) [112]. Its properties and strategies for extraction from hops and brewery residues were recently described [113]. The involvement of xanthohumol in human malignancies is widely documented [114]. Xanthohumol dose-dependently reduces the growth of human MCF-7 BC cells and is cytotoxic to the MCF-7 cell line at a concentration of 100 µM [115]. It was demonstrated to inhibit cellular proliferation in MDA-MB231 BC cell lines through an intrinsic mitochondrial-dependent pathway [116]. It also inhibits the invasion of triple-negative and hormone-dependent BCs [117] and interacts with efflux transporter BC resistance protein (BCRP)/adenosine triphosphate binding cassette G2 (ABCG2) [118]. Xanthohumol is radiosensitizing in doxorubicin-resistant MCF-7 cells. The radio-sensitizing effect of xanthohumol is likely mediated by STAT3 and EGFR suppression in doxorubicin-resistant MCF-7 human BC cells [119]. Moreover, xanthohumol acts on oxidative estrogen metabolism by inhibiting aromatase [120,121]. Xanthohumol suppresses estrogen signaling in BC through the inhibition of BIG3-PHB2 interactions [122,123,124]. To date, no side effects affecting any organ or body district have been described. A recent review summarized the effects of xanthohumol supplementation in diverse studies on humans [125]. None of these studies on xanthohumol pharmacokinetics described in the review led to adverse effects. A placebo-controlled clinical trial (the XMaS Trial) on the safety and tolerability of xanthohumol showed that a daily intake of 24 mg xanthohumol over 8 weeks is safe and well tolerated with all clinical biomarkers and anthropometrics being unaffected and/or staying within the clinically normal reference range [126]. The doses indicated by ESCOP are 30 mg/kg/die [18].

### 7.2. Citrus Bioflavonoids

Citrus bioflavonoids are flavonoids present in all types of citrus fruits, including oranges, grapefruits, lemons and limes, and constitute the pigments responsible for the typical yellow-orange color of such fruits. Citrus fruits contain high concentrations of several classes of phenols, including numerous hydroxycinnamates, flavonoid glycosides and polymethoxylated flavones. The latter group of compounds occurred without glycosidic bonds and was shown to inhibit the proliferation of several cancer cell lines, including that of BC [127]. The anti-estrogenic and anti-aromatase activities of citrus peel major compounds in BC were reported [128]. Citrus bioflavonoids include rutin, which is a quercetin glucoside [129], and hesperidin [130]. Until now, no side effects affecting any organ or body district have been reported. Hesperidin is safely administered, even with pregnancy, with no detected side effects and noncumulative properties [131]. Among the secondary metabolites of great benefit to humans are flavonoids, which are polyphenolic compounds widely distributed in the plant kingdom and, in particular, synthesized by plants in response to microbial infections. Citrus flavonoids are effective inhibitors of both human MDA-MB-435 breast carcinoma cells negative for estrogen receptor and MCF-7 positive for estrogen receptor in vitro [132,133]. Moreover, 1:1 combinations of flavonoids with tocotrienols and/or tamoxifen inhibit cell proliferation more effectively than single compounds. This synergy may be due to the fact that the compounds are exerting their inhibitory effects with different mechanisms [134]. Furthermore, citrus bioflavonoids are potentially able to inhibit dormant neoplastic stem cells [135]. 

## 8. Myrrh

Myrrh is an aromatic gum resin extracted from a tree or shrub of the genus Commiphora in the *Burseraceae* family. It is studied for its anti-cancer, anti-inflammatory and antibacterial activities [136,137,138]. Myrrh was approved by the FDA in 1992 as a safe additive and is included in the list of generally recognized as safe (GRAS) substances and included in the plant and part list by the Council of Europe [139]. Three species of myrrh are known, namely *Commiphora myrrh*, *Commifora molmol* and *Commiphora wightii*. They are all studied for their anticancerogenic potential [140,141]. Specifically, *Commiphora molmol* (oleo-resin) was studied in vivo at 150 and 500 mg/kg in mice bearing Ehrlich’s ascites and normal carcinoma cells; the viability of these cells revealed a cytotoxic and antitumor activity similar to that of the drug cyclophosphamide, as evaluated in terms of the changes of liver cells and bone marrow on the basis of the frequency of micronuclei and polychromatic and normochromatic cells. It also possesses chemosensitizing activity by inhibiting glycoprotein-P, which is an ABC pump responsible for transporting various substances across the plasma membrane, mitochondrial membrane, endoplasmic reticulum and peroxisomes. It inhibits MMP-P through the inhibition of NF-kB and AP1, which are responsible for its activation. Moreover, it inhibits proliferation by arresting the cell cycle in the S phase [142]. The significant therapeutic benefits of myrrh have been attributed to the activity of its diverse metabolites, which were recently identified by Suliman et al. [143]; in this work, the authors analyzed the action in vitro of active metabolites from a myrrh resin methanolic extract against leukemia and BC cell lines. Particularly, for BC, the myrrh extract showed moderate cytotoxic activity against the KAIMRC1 BC cell line. Furthermore, the oleo-gum resin of *Commiphora myrrh* was recently studied in vitro against HeLa BC cell lines, showing that myrrh triterpenes showed cytotoxicity against these cell lines [144]. The effects of *Commiphora myrrh* in triple-negative BC (TNBC) were recently investigated by studying its component Z-guggulsterone. It inhibits the proliferation of the TNBC cell lines MDA-MB-468 and BT-549 in vitro. In vivo studies demonstrated that it inhibits TNBC progression via the p53/cyclin B1 (CCNB1)/polo-like Kinase 1 (PLK1) pathway, inducing cell cycle arrest and apoptosis [145]. In a recent study, the synergism of myrrh with paclitaxel was demonstrated. Specifically, paclitaxel was incorporated in an effective nanocarrier formulation, namely, a nanoemulsion formulated by using myrrh essential oil and polyethylene glycol distearoylphosphatidylethanolamine (PEG-DSPE) leading to enhanced cytotoxicity against MDA-MB231 BC cell lines [146]. To our knowledge, there is no NCT regarding the use of myrrh in the treatment of BC. Messina et al. described a randomized study of complementary supportive medicine with Aloe arborescens vs. Aloe + myrrh using doses of 10 mL twice/day in metastatic solid tumor patients who did not respond to the standard anticancer therapies. Both therapies induced disease control in a percentage that was significantly higher for patients treated with Aloe plus myrrh than that achieved in patients treated with Aloe alone [147]. Myrrh can be used in 150–500 mg/kg doses [18].

## 9. Sulforaphane

Sulforaphane (1-isothiocyanate-4-(methylsulfinyl)butane) is a chemical compound belonging to the class of isothiocyanates (ITCs) and naturally present as a metabolite of the phytochemical glucoraphanin in plants belonging to the cruciferous group (broccoli, cabbage, Brussels sprouts, cauliflower, turnip greens). Numerous in vitro, in vivo and clinical studies were reported on its anticancer activity [148]. Several studies documented the cancer-preventive activity of a significant number of ITCs, the majority of which occur in plants, especially in cruciferous vegetables. Its potential as an anticancer agent was recently reviewed [149]. Recently, the molecular mechanisms related to sulforaphane as an adjuvant treatment in BC were described [150]. Sulforaphane is capable of inhibiting histone deacetylase through global demethylation, modulating microRNAs by demethylating the first 5 GPGs, affecting microtubules and inhibiting the expression of MMPs [151]. Sulforaphane can activate the Erythroid Nuclear Factor (Nrf-2), which is an important transcription factor in controlling the antioxidant system, which is silenced during mammary tumorigenesis [152]. It allows for the decrease of survivin and inhibits selectin E, which is responsible for bone micrometastases, which can remain dormant for more than ten years [153]. It sensitizes HER-2 BC cells positive to chemotherapy [154] and inhibits the Hedgehog signaling pathway in stem cells for the self-maintenance of cancer cells [68]. Sulforaphane is generally safe and well-tolerated at low doses [155]. During therapy with doxacicline, sulforaphane at a dosage of 4 mg/kg as an adjuvant helps to reduce cardiotoxicity by enhancing mitochondrial activity. Moreover, sulforaphane plus doxacicline showed significantly greater tumor regression than sulforaphane or doxacicline alone [156]. For high doses, the safety of sulforaphane is under study: some authors report that it can lead to toxicity and adverse effects [157]. It is used in colorectal cancer, even though evidence about its safety in these patients are still lacking [158]. Doses indicated by ESCOP range from 600 to 1200 mg/die [18].

## 10. Vitamin D

Vitamin D exists in two forms: vitamin D2 or ergocalciferol, of plant origin, and vitamin D3 or cholecalciferol, derived from cholesterol and produced directly by the organism. It is a prohormone and needs to be hydroxylated at the liver and kidney levels. It is mainly synthesized during sun exposure, while dietary intake provides about 10–15% of it. Elevated blood levels of 5-hydroxyvitamin D (25OHD), or vitamin D, are directly correlated with better overall survival in patients with BC, providing compelling observational evidence of an inverse correlation between vitamin D levels, mortality and BC progression risk [159]. Vitamin D3 was shown to reduce the incidence of BC in humans and induce apoptosis and cell cycle arrest of various tumor cells. In 2001, Palmer et al. demonstrated that vitamin D3 promoted the differentiation of colon carcinoma cells by inducing E-cadherin expression and inhibiting β-catenin signaling. The vitamin D ligand-activated receptor competed with TCF-4 for β-catenin binding, thus reducing the levels of c-Myc, peroxisome proliferator-activated receptor, TCF-1 and CD44.

## 11. Mushrooms

Mushrooms belong to the astounding dominion of Fungi and are known as a macrofungus. Significant health benefits of mushrooms were reported worldwide, including anticancer, antioxidant and radical scavenging effects. Mushrooms primarily contain low- and high-molecular-weight polysaccharides, fatty acids, lectins and glucans that are responsible for their therapeutic action [160]. Their activity is related to CD8+ lymphocytes and CTLs, which are the main modulators in the elimination of tumor cells. Their use may be indicated, especially when anomalies are found in the lymphocyte subpopulation responsible for eliminating cancer cells [161,162,163,164].

## 12. Acetylsalicylic Acid

Acetylsalicylic acid (ASA), better known as aspirin, is a chemical modification of salicin, which was originally derived from the bark of the willow tree. When salicin is dissolved in water, it is acidic, and a saturated solution of it has a pH of 2.4, which is why it was renamed salicylic acid. The use of low doses of acetylsalicylic acid is associated with a significant increase in survival in neoplastic disease [165,166,167,168]. Recently, its effects on BC epithelial cells was studied [169]. 

## 13. Metformin

Metformin is a hypoglycemic drug belonging to the group of biguanides that are derived from the plant *Galega officinalis* L. Metformin reduces the expression of genes that encode for IGF-1 and the expression of IGF-1R receptors [170]. It inhibits the transition of malignant epithelial cells to mesenchymal through the blockade of TGF-beta1, reducing metastasis and their progression [171,172,173]. It upregulates the tumor suppressor microRNA 34A [174]. It can be used as a chemopreventive for primary prevention in association with anti-hormonal drugs in BC [175,176], as it acts metabolically and epigenetically on cancer cells [177,178]. It can be used as a sensitizer during gemcitabine, 5-fluorouracil, epirubicin and cyclophosphamide chemotherapies [179,180], as well as during radiotherapy, as it sensitizes cells to radiotherapeutic treatment [181,182].

## 14. Clinical Studies 

In this section, the recent clinical studies (National Clinical Trials (NCTs)) on BC with the involvement of the natural products described are summarized (Table 2).

## 15. Conclusions

BC is the most frequent cancer among women. Numerous mechanisms are related to the onset and progression of BC. In BC, inflammation and cancer influence each other, and the inflammatory component precedes the development of the neoplasm, but slowly increasing and prolonged inflammation related to the neoplastic phenomenon also favors its growth. Peritumoral inflammation plays an important role in carcinogenesis, and HSPs play an active part in malignant transformation and progression by allowing the expression of new malignant traits through an accumulation of altered oncoproteins. In this context, natural products play a crucial role. Curcumin; EGCG; I3C; artesunate; ginger; flavonoids and bioflavonoids, such as xanthohumol and citrus bioflavonoids; myrrh; sulforaphane; vitamin D; medicinal mushrooms; acetylsalicylic acid; and metformin were analyzed in detail in order to underline their importance in the onset, progression and/or treatment of BC. The clinical studies on these compounds in BCs were detailed. The integration with these products can be implemented both during therapy in the attack phase, and subsequently to classic protocols. It was shown that some natural substances, through their integration into the classic protocol, produce a synergistic effect with the latter through the stabilization of altered signaling pathways, inhibition of inflammatory factors and increase in immune capabilities. Moreover, some aspects regarding the safety and toxicological involvement of these compounds were analyzed. It must be considered that during the chemotherapy and radiotherapy phase, the dosages taken as integration should be congruent with those that can be taken in the follow-up phase. Some more clinical studies are needed in order to assess the importance of these compounds.

## Figures and Tables

**Table 1 cancers-15-02981-t001:** Natural products used in the treatment of BC.

Structure	Name
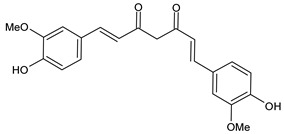	Curcumin
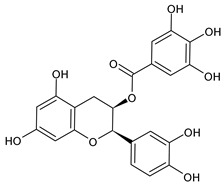	Epigallocatechin gallate (EGCG)
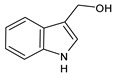	Indole-3-carbinol (I3C)
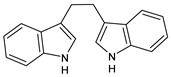	Diindolylmethane
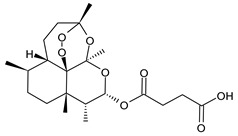	Artesunate
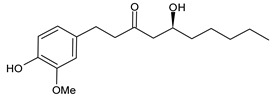	6-Gingerol
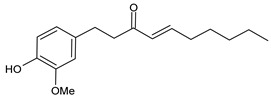	6-Shogaol (6SG)
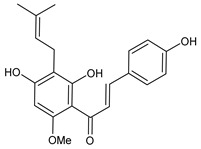	Xanthohumol
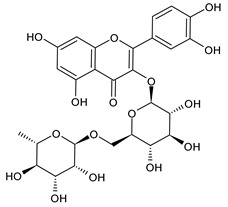	Rutin
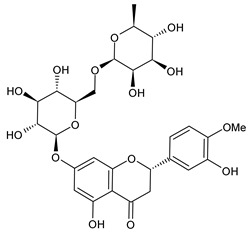	Hesperidin
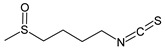	Sulforaphane
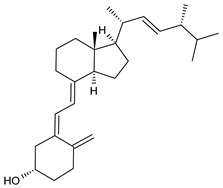	Vitamin D2
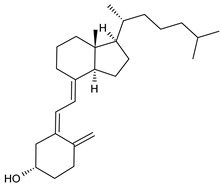	Vitamin D3
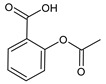	Acetylsalicylic acid (ASA)
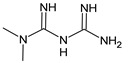	Metformin

**Table 2 cancers-15-02981-t002:** Clinical studies on natural products involving BC.

Natural Product	Name of the Clinical Trial	NCT Number	Doses	Phase	Trial Status
Curcumin	“Curcumin” in Combination With Chemotherapy in Advanced Breast Cancer	NCT03072992	300 mg i.v.	Phase 2	Completed
Curcumin	Phase II Study of Curcumin vs. Placebo for Chemotherapy-Treated Breast Cancer Patients Undergoing Radiotherapy	NCT01740323	500 mg BID MERIVA corresponding to ~90 mg of curcumin	Phase 2	Completed
Epigallocatechin-3-gallate (EGCG)	Study of Topically Applied Green Tea Extract for Radio Dermatitis and Radiation Mucositis	NCT01481818	Application of 0.01~0.05 mL/cm^2^ 3 times a day to the area under treatment during radiotherapy	Phase 2	Unknown
Indole-3-Carbinol	Indole-3-Carbinol in Preventing Breast Cancer in Nonsmoking Women Who Are at High Risk For Breast Cancer (clinical trial: NCT00033345)	NCT00033345	400–800 mg pill taken daily	Phase 1	Completed
Artesunate	Study of Artesunate in Metastatic Breast Cancer	NCT00764036	Add-on therapy with daily single oral doses: 100, 150 or 200 mg	Phase 1	Completed
Ginger (6-gingerol and 6-shogaol)	Ginger in Treating Nausea in Patients Receiving Chemotherapy for Cancer	NCT00040742	0.5–1.5 g oral high-dose ginger twice daily	Phase 3	Completed
Xanthohumol	Xanthohumol and Prevention of DNA Damage	NCT02432651	6–24 mg xanthohumol per day	Phase 1	Completed
Bioflavonoids	Defined Green Tea Catechin Extract in Treating Women With Hormone Receptor Negative Stage I-III Breast Cancer	NCT00516243	Green tea catechin extract PO BID (amount not given)	Phase 1	Completed
Bioflavonoids	Green Tea and Reduction of Breast Cancer Risk	NCT00917735	Two green tea extract capsules, each containing 80.7% total catechins (51.7% EGCG) twice daily	Phase 2	Completed
Bioflavonoids	Disposition of Dietary Polyphenols and Methylxanthines in Mammary Tissues From Breast Cancer Patients (POLYSEN)	NCT03482401	3 capsules/day (474 mg phenolics/day)	NA	Completed
Sulforaphane	Study to Evaluate the Effect of Sulforaphane in Broccoli Sprout Extract on Breast Tissue	NCT00982319	100 µmol of sulforaphane (dissolvable) in Broccoli sprout extract	Phase 2	Completed
SFX-01 (Sulforaphane + alpha-cyclodextrin)	SFX-01 in the Treatment and Evaluation of Metastatic Breast Cancer (STEM)	NCT02970682	SFX-01, provided as 300 mg capsules, one to be taken twice daily	Phase 2	Completed
Vitamin D	Vitamin D can Increase Pathological Response of the Breast Cancer Patients Treated with Neoadjuvant Therapy	NCT03986268	50,000 IU weekly	NA	Unknown
Mushrooms	White Button Mushroom Extract in Preventing the Recurrence of Breast Cancer in Postmenopausal Breast Cancer Survivors	NCT00709020	Escalating doses: 5 g/day, then 8 g/day, then 10 g/day, then 13 g/day	Phase 1	Completed
Acetylsalicylic acid	Aspirin in Preventing Recurrence of Cancer in Patients with HER2 Negative Stage II-III Breast Cancer After Chemotherapy, Surgery, and/or Radiation Therapy	NCT02927249	300 mg daily	Phase 3	Terminated
Acetylsalicylic acid	Low Dose Chemotherapy With Aspirin in Patients With Breast Cancer After Neoadjuvant Chemotherapy	NCT01612247	325 mg PO daily	NA	Unknown
Metformin Hydrocloride	Metformin Hydrochloride vs. Placebo in Overweight or Obese Patients at Elevated Risk for Breast Cancer	NCT01793948	850 mg PO BID	Early phase 1	Completed
Metformin	Metformin Hydrochloride in Preventing Breast Cancer in Patients With Atypical Hyperplasia or in Situ Breast Cancer	NCT01905046	850 mg PO BID	Phase 3	Active, not recruiting

BID—twice-daily; NA—not applicable; PO—orally.

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
