# Peer review of "Natural Products for the Prevention, Treatment and Progression of Breast Cancer"

_cancers, 2023, doi:10.3390/cancers15112981_

Round 1

Reviewer 1 Report

Breast cancer (BC) is the leading cancer that affects women. Treatment includes combinations of radiotherapy, chemotherapy, and surgery, the efficacy of which can be greatly enhanced or compromised by consuming bioactive plant substances, such as polyphenols or synthetic equivalents. This review summarises the current information on the natural products and synthetic equivalents most widely used for the prevention, treatment, and progression of BC.

The authors have thoroughly overviewed the diverse molecular pathways and mechanisms disrupted in the onset and development of BC and how bioactive factors could ameliorate deleterious changes. However, the range and complexity of the information presented make it difficult to tease out what findings are from clinical settings and which are from studies in vitro. All the data is relevant, but a table summarising the factors tested in a clinical study, the doses/dosing regime, and overall efficacy should be added to the text. This will allow the reader to better focus on and interpret the key aspects of information.

Pg 2 para 1 ln 9-18     This text touches upon a complex set of positive and negative reactions. Expand this text and provide more detail of processes with regard to cancer.

Pg 2 para 2 (2. Molecular bases of BC)         basis?

Pg 2 para 2 ln 1-6       In healthy responses, initial activation of NFκB is subsequently downregulated by suppressors, such as TNF alpha-induced protein 3 [TNFAIP3; A20]. Any indications in the literature of failure of A20 or other suppressor pathways in BC?

Pg 2 para 2 ln 20-22   Expand on this.

Pg 2 para 2 ln 29         ‘dialogue’. Suggest ‘interact and crosstalk’.

Pg 3 para 1 ln 2           ‘lose’. Suggest ‘block or impair ‘.

3-13     Highlight which has been tested in clinical trials.

Minor changes to English are needed. See the above notes.

Author Response

REVIEWER 1:

The authors have thoroughly overviewed the diverse molecular pathways and mechanisms disrupted in the onset and development of BC and how bioactive factors could ameliorate deleterious changes. However, the range and complexity of the information presented make it difficult to tease out what findings are from clinical settings and which are from studies in vitro. All the data is relevant, but a table summarising the factors tested in a clinical study, the doses/dosing regime, and overall efficacy should be added to the text. This will allow the reader to better focus on and interpret the key aspects of information.

We thank the reviewer for these observations. Actually, there are numerous preclinical and clinical studies on the described compounds. References and sentences referring to preclinical studies were added in the text. Commonly used doses were added in the text with the appropriate reference and/or taken from ESCOP Monographs, 2/e. ESCOP, The European Scientific Cooperative on Phythoterapy, Argyle House, Gandy Street, Exter, EX4 3LS, Devon, Devon, United Kingdom. However, given that these natural substances are prescribed in addition to traditional oncological therapies in clinical studies, we thought to focus on the clinical studies (NCT) of these compounds in the treatment of BC patients. Thus, we added paragraph 14 and Table 2 in which we reported name, number, phase and status of the NCTs and the relative doses used in the trial.

Pg 2 para 1 ln 9-18   This text touches upon a complex set of positive and negative reactions. Expand this text and provide more detail of processes with regard to cancer.

We thank the reviewer for this observation. Actually, the aim of the review was to focus on natural products. More details on studies on BCs have been recently described in another paper of our group. It was added in the text as reference 2.

Pg 2 para 2 (2. Molecular bases of BC) basis?

We thank the reviewer for this correction. Bases was changed into basis.

Pg 2 para 2 ln 1-6 In healthy responses, initial activation of NFκB is subsequently downregulated by suppressors, such as TNF alpha-induced protein 3 [TNFAIP3; A20]. Any indications in the literature of failure of A20 or other suppressor pathways in BC?

We thank the reviewer for this suggestion. A sentence was added in the text (in Paragraph 2. Molecular basis of BC) explaining this topic.

Pg 2 para 2 ln 20-22 Expand on this.

We thank the reviewer for this suggestion. A sentence was added in the text (in Paragraph 2. Molecular basis of BC) explaining this topic.

Pg 2 para 2 ln 29 ‘dialogue’. Suggest ‘interact and crosstalk’.

We thank the reviewer for this suggestion. Dialogue was changed into cross-talk.

Pg 3 para 1 ln 2 ‘lose’. Suggest ‘block or impair ‘.

We thank the reviewer for this suggestion. Lose was changed into impair.

3-13 Highlight which has been tested in clinical trials.

We thank the reviewer for this suggestion. Table 2 was added in the text.

Reviewer 2 Report

The review provides a good overview of the current understanding of breast cancer pathophysiology and the potential role of natural products as adjuvants in its prevention and treatment. However, there are some points that could be improved. Some specific comments include:

•             The review's goal is not clearly stated in the abstract. It would be beneficial to clarify the purpose of the review explicitly at the beginning of the abstract.

•             To lead the review, the introduction could benefit from a more defined and targeted research question or objective.

•             The pathophysiology part may be more organized and clearer. Without a clear pattern or flow, the paragraph discusses several elements (inflammation, peritumoral inflammation, HSPs, MMPs, neoplastic stem cells, immunological modulation, microRNAs).

•             The section on neoplastic stem cells may be simplified for improved readability. In addition, it would be useful to clarify why neoplastic stem cells are significant in breast cancer and how they link to the disease's pathophysiology.

•             It would be beneficial to include more particular examples of natural items that have been investigated as adjuvants in breast cancer prevention and treatment. Currently, the section on natural products is vague and lacks precise information about which natural items have been investigated.

•             Artesunate

•    The paragraph lacks context and does not clearly state the relevance of artesunate in cancer treatment. Consider providing more background information.

•    The sentence "It is able to regulate microRNA-34a, which is in turn regulated by P53" is unclear. Please provide more information on how artesunate regulates microRNA-34a.

•    The section would benefit from providing more detail on the mechanism of action of artesunate.

Ginger and its Constituents

Gingerol

•    Consider adding more information on the source and extraction of 6-gingerol.

•    The sentence "It sensitizes cell death induced by TRIAL by apoptosis" is unclear. Please rephrase or provide more information.

•    The section could benefit from providing more information on the potential toxicity of 6-gingerol.

Flavonoids and Bioflavonoids

Xanthohumol

•    The paragraph might benefit from further context regarding Xanthohumol and its significance in cancer therapy.

•    The sentence "The activity of xanthohumol has proved to be higher than cisplatin" is unclear. Please provide more information on the basis for this comparison.

•    The paragraph could benefit from providing more information on the potential toxicity of Xanthohumol.

Citrus Bioflavonoids

•    The paragraph lacks context and does not describe the significance of citrus bioflavonoids in cancer treatment in a straightforward manner. Consider including more context details.

•    Regarding the potential toxicity of citrus bioflavonoids, this paragraph could benefit from additional information.

Myrrh

•    The background is not sufficient, and the relation of myrrh to cancer treatment is not stated effectively. Consider offering further context. The paragraph could benefit from providing more detail on the mechanism of action of myrrh.

Sulforaphane

•    The sentence "It is found in various plant species, especially in cruciferous plants" is unclear. Please provide more information on which plant species are known to contain sulforaphane.

•    More information regarding the potential toxicity of sulforaphane could be included in this section.

•    The section lacks an introduction that provides context for the importance of polyphenols in cancer prevention and treatment.

•    The references in the section are not consistent in style.

•    The section could be divided into more digestible paragraphs to make it easier to read and comprehend.

•    The section contains technical terms and acronyms that may not be familiar to all readers. Therefore, the first time each term or acronym is used, it should be defined.

•    The section could benefit from using subheadings to organize the different polyphenols.

•    The section could use more information about the mechanisms of action of the different polyphenols.

•    The section could benefit from discussing the clinical evidence for the effectiveness of the different polyphenols in preventing or treating cancer.

•    The section could use more information about the safety and potential side effects of the different polyphenols.

Author Response

REVIEWER 2:

The review's goal is not clearly stated in the abstract. It would be beneficial to clarify the purpose of the review explicitly at the beginning of the abstract.

We thank the referee for this suggestion. The aim of the review was shifted from the end to the beginning of the abstract and it was enlarged. The purpose of this review is to clarify how certain natural products may play a role in the prevention, treatment and progression of breast cancer.

To lead the review, the introduction could benefit from a more defined and targeted research question or objective.

We thank the referee for this suggestion. The aim of the research was to further explore natural products in nature that interfered in the onset, maintenance and progression of BC since this disease is influenced by multiple pathophysiological and signaling aspects.

The pathophysiology part may be more organized and clearer. Without a clear pattern or flow, the paragraph discusses several elements (inflammation, peritumoral inflammation, HSPs, MMPs, neoplastic stem cells, immunological modulation, microRNAs).

We thank the referee for this suggestion. A sentence was added in the text explaining this topic.

The section on neoplastic stem cells may be simplified for improved readability. In addition, it would be useful to clarify why neoplastic stem cells are significant in breast cancer and how they link to the disease's pathophysiology.

We thank the referee for this suggestion. A little paragraph regarding neoplastic stem cells was added in the text (Paragraph 2. Molecular basis of BC).  

It would be beneficial to include more particular examples of natural items that have been investigated as adjuvants in breast cancer prevention and treatment. Currently, the section on natural products is vague and lacks precise information about which natural items have been investigated.

We thank the reviewer for this suggestion. We add some sentences at the end of introduction regarding other natural products used and the reason for which we focused only on some of them.

Artesunate

  •   The paragraph lacks context and does not clearly state the relevance of artesunate in cancer treatment. Consider providing more background information.
  •   The sentence "It is able to regulate microRNA-34a, which is in turn regulated by P53" is unclear. Please provide more information on how artesunate regulates microRNA-34a.
  •   The section would benefit from providing more detail on the mechanism of action of artesunate.

We thank the referee for these suggestions. We add some sentences in the text to better clarify the involvement of artesunate in BC.

Gingerol

  •   Consider adding more information on the source and extraction of 6-gingerol.

We thank the referee for this suggestion. Some sentences and references were added in the text.

  •   The sentence "It sensitizes cell death induced by TRIAL by apoptosis" is unclear. Please rephrase or provide more information. It sensitizes cell death induced by TRAIL (Tumor necrosis factor (TNF)-related apoptosis-inducing ligand) by apoptosis

We thank the referee for these suggestions. We add some sentences in the text regarding toxicity. Moreover, Tumor necrosis factor (TNF)-related apoptosis-inducing ligand was added in the text.  

  •   The section could benefit from providing more information on the potential toxicity of 6-gingerol.

We thank the referee for this suggestion. it was added.

Xanthohumol

  •   The paragraph might benefit from further context regarding Xanthohumol and its significance in cancer therapy.

We thank the referee for these suggestions. We add some sentences in the text.

  •   The sentence “The activity of xanthohumol has proved to be higher than cisplatin” is unclear. Please provide more information on the basis for this comparison.

We thank the referee for this observation. It was not clear and correct. The sentence was deleted.

  •   The paragraph could benefit from providing more information on the potential toxicity of Xanthohumol.

We thank the referee for these suggestions. We add some sentences in the text regarding Xanthohumol toxicity.

Citrus Bioflavonoids

  •   The paragraph lacks context and does not describe the significance of citrus bioflavonoids in cancer treatment in a straightforward manner. Consider including more context details.
  •   Regarding the potential toxicity of citrus bioflavonoids, this paragraph could benefit from additional information.

We thank the referee for these suggestions. We add some sentences and references in the text.

Myrrh

  •   The background is not sufficient, and the relation of myrrh to cancer treatment is not stated effectively. Consider offering further context. The paragraph could benefit from providing more detail on the mechanism of action of myrrh.

We thank the referee for this suggestion. The paragraph was enlarged with some sentences and references regarding studies on the different existing species of myrrh. The recent studies show its very interesting potential as anticancer. We didn’t find any NCT regarding the use of myrrh in BC, thus we cited a clinical study of myrrh only in the text, but not in Table 2 (Messina et al. Med. Clin. Sci. 2022, 4, 1-3).

Sulforaphane

  •   The sentence "It is found in various plant species, especially in cruciferous plants" is unclear. Please provide more information on which plant species are known to contain sulforaphane.

We thank the reviewer for this observation. Sulforaphane is a chemical compound belonging to the isothiocyanate family and naturally present in plants belonging to the cruciferous group (broccoli, cabbage, Brussels sprouts, cauliflower, turnip greens).

  •   More information regarding the potential toxicity of sulforaphane could be included in this section.

We thank the reviewer for this observation.  To date, no side effects affecting any organ or body district have been identified. So sulforaphane can be considered an absolutely safe natural substance.

  •   The section lacks an introduction that provides context for the importance of polyphenols in cancer prevention and treatment.

We thank the reviewer for this observation. A sentence was added in the text.

  •   The references in the section are not consistent in style.

We thank the reviewer for this observation. References were corrected.

  •   The section could be divided into more digestible paragraphs to make it easier to read and comprehend.
  •   The section could benefit from using subheadings to organize the different polyphenols.

We thank the reviewer for these observations. However, some polyphenols were in the paragraph of flavonoids, as they belong to this class. In the polyphenols we cited only curcumin and EGCG.

  •   The section contains technical terms and acronyms that may not be familiar to all readers. Therefore, the first time each term or acronym is used, it should be defined.

We thank the reviewer for this suggestion. All acronyms were checked, corrected and when mentioned for the first time the name was added. Moreover, abbreviations for all the acronyms used in the text were added at the end of the paper.

  •   The section could use more information about the mechanisms of action of the different polyphenols.

We thank the reviewer for these suggestions. The mechanism of action of polyphenols have been widely described in the literature. We mentioned the modulation of microRNAs, MAPK signaling and epigenetics along with some appropriate references. Moreover, we added that polyphenols may also affect overcome chemoresistance in cancer cells, as recently reported.

  •   The section could benefit from discussing the clinical evidence for the effectiveness of the different polyphenols in preventing or treating cancer.

We thank the reviewer for these suggestions. Some sentences were added in the text.

  •   The section could use more information about the safety and potential side effects of the different polyphenols.

We thank the reviewer for these suggestions. A sentence and a reference were added in the text. Clinical evidence for the effectiveness of the different polyphenols in preventing or treating cancer can be deduced from table 2 regarding clinical studies that was added in the text.

Round 2

Reviewer 2 Report

No further comments!